# The Role of Endocrine Disruption Chemical-Regulated Aryl Hydrocarbon Receptor Activity in the Pathogenesis of Pancreatic Diseases and Cancer

**DOI:** 10.3390/ijms25073818

**Published:** 2024-03-29

**Authors:** Kyounghyun Kim

**Affiliations:** Department of Pharmacology and Toxicology, College of Medicine, University of Arkansas Medical Sciences, Little Rock, AR 72225, USA; kkim@uams.edu; Tel.: +1-501-526-6000 (ext. 24306); Fax: +1-501-686-5510

**Keywords:** endocrine disrupting chemicals, environmental toxicants, aryl hydrocarbon receptor, pancreas, pancreatitis, pancreatic injury, pancreatic cancer

## Abstract

The aryl hydrocarbon receptor (AHR) serves as a ligand-activated transcription factor crucial for regulating fundamental cellular and molecular processes, such as xenobiotic metabolism, immune responses, and cancer development. Notably, a spectrum of endocrine-disrupting chemicals (EDCs) act as agonists or antagonists of AHR, leading to the dysregulation of pivotal cellular and molecular processes and endocrine system disruption. Accumulating evidence suggests a correlation between EDC exposure and the onset of diverse pancreatic diseases, including diabetes, pancreatitis, and pancreatic cancer. Despite this association, the mechanistic role of AHR as a linchpin molecule in EDC exposure-related pathogenesis of pancreatic diseases and cancer remains unexplored. This review comprehensively examines the involvement of AHR in EDC exposure-mediated regulation of pancreatic pathogenesis, emphasizing AHR as a potential therapeutic target for the pathogenesis of pancreatic diseases and cancer.

## 1. Introduction

EDCs represent exogenous substances or mixtures that disrupt endocrine system function, leading to adverse effects in organisms, their progeny, or specific populations. These substances dysregulate the endocrine system, influencing hormone production, storage, and secretion, contributing to various detrimental effects on human health. EDCs can mimic, block, or interfere with the body’s hormone functions. Consequently, exposure to EDCs is associated with disruptions in sperm count, fertility, reproductive organs, endometriosis, puberty, cardiovascular function, immune response, nervous system activity, respiratory function, and metabolism. Ultimately, these disruptions may contribute to the development of various human diseases and cancers [1,2,3].

EDCs comprise a broad range of exogenous substances, including bisphenols [4], phthalates [5], organotin [6], pesticides [7], polychlorinated dibenzo-p-dioxins (PCBs) [8], dioxin-like compounds [9], polyaromatic hydrocarbons (PAHs) [10], flame retardants [11], and alkylphenols [12]. Exposure to EDCs is widespread in daily life through various products such as cosmetics, food and beverage packaging, toys, and carpets [13,14,15]. Naturally occurring EDCs, like phytoestrogens such as genistein and daidzein, contribute to this diverse array [16]. Certain heavy metals, including arsenic, chromium, or cadmium, can also function as endocrine disruptors [17,18,19]. Exposure routes to EDCs are diverse, involving ingestion, water consumption, inhalation, and dermal contact. Notably, many EDCs can act as agonists or antagonists of the aryl hydrocarbon receptor (AHR), a ligand-activated transcription factor crucial for environmental sensing and xenobiotic metabolism [20,21,22,23,24].

## 2. Aryl Hydrocarbon Receptor (AHR)

The aryl hydrocarbon receptor (AHR) was initially identified as a cytoplasmic receptor with a high affinity for 2,3,7,8-tetrachlorodibenzo-p-dioxin (TCDD) [25]. The harmful health effects resulting from accidental TCDD exposure were first reported in Nitro, West Virginia, in 1949 [26]. TCDD gained notoriety as a contaminant in Agent Orange, an herbicide and defoliant mixture widely used during the Vietnam War, composed of N-butyl esters of 2,4-dichlorophenoxyacetic (2,4-D) and 2,4,5-trichlorophenoxyacetic (2,4,5-T) acids. Veterans with high exposure to Agent Orange have exhibited an increased incidence of cancer and congenital disabilities in their children [27]. TCDD, the most potent and extensively studied dioxin, serves as a prototype for dioxins that function as AHR agonists [28].

Basically, AHR functions as a ligand-activated transcription factor integral to cellular homeostasis, governing various physiological and pathological processes, including xenobiotic detoxification, metabolism, cardiovascular regulation, immunomodulation, and cancer development [29,30,31]. In its unliganded state, AHR forms an inactive complex, engaging with two heat shock protein 90 (HSP90) [32,33], AHR interacting protein (AIP) [34,35], and prostaglandin E synthase 3 (PTGES3 or p23) [36]. This interaction serves to maintain AHR stability, conformation, and cytoplasmic localization. AIP, in particular, safeguards AHR from ubiquitylation-induced degradation while contributing to AHR folding and stability through direct interactions with HSP90 and AHR [37].

Upon ligand binding, the aryl hydrocarbon receptor (AHR) undergoes nuclear translocation, exposing its nuclear localization signal (NLS). Within the nucleus, AHR forms a complex with the AHR nuclear translocator (ARNT), also known as Hypoxia Inducible Factor 1 Beta (HIF1β). This AHR/ARNT heterodimer binds specifically to DNA sequences termed Xenobiotic Response Elements (XREs), with a consensus sequence of 5′-TNGCGTG-3′. These XREs are situated in the promoter regions of downstream target genes, initiating the activation of gene expression (Figure 1). The interactions between AHR-ARNT-XRE lead to the induction of downstream target genes, including phase I detoxification enzymes such as CYP1A1 and CYP1B1, as well as phase II detoxification enzymes like UGT1A1 and UGT1A6 [36,37,38,39]. Following the activation of AHR signaling, the aryl hydrocarbon receptor repressor (AHRR) protein is induced, exerting inhibitory effects on AHR signaling activation. AHRR competes with ARNT to bind to AHR, constituting a negative feedback mechanism [40]. The resultant gene expressions have diverse physiological, pathological, and toxicological implications in various human diseases and cancers. This canonical AHR pathway activation, mediated through binding the AHR/ARNT complex to XRE, is recognized as the primary mechanism of AHR signaling.

In contrast to the canonical AHR pathway, which relies on AHR/ARNT/XRE interactions, activating the non-canonical AHR pathway occurs through interactions with other transcription factors. These factors include Kruppel-like factor 6 (KLF6), Estrogen Receptor α (ERα), and a member of the NF-κB family such as RelB. Independent of AHR/ARNT complex formation, AHR forms a heterodimer with KLF6 and binds to non-consensus XRE, inducing gene expression [41]. Ligand-activated AHR inhibits gene expression responses to the estrogen/ERα complex [42]. The interaction between ligand-bound AHR and RelB regulates IL8 expression, which is crucial in developing chronic inflammatory diseases [43] (Figure 1). Additionally, the non-canonical AHR pathway involves PKA (cAMP-dependent protein kinase)-mediated AHR activation in a ligand-independent manner [44]. The role of the non-canonical AHR pathway in various human diseases, including pancreatic diseases and cancer, remains largely unexplored.

## 3. AHR Structure and Its Interactions with Various Ligands

AHR belongs to the basic helix-loop-helix/per-ARNT-sim (bHLH/PAS) superfamily, characterized by three functional structural domains: a bHLH domain responsible for DNA binding, two PAS structural domains (A and B) facilitating dimerization with ARNT and ligand binding, and a transactivating domain for gene expression (Figure 2A). PAS domains function as ubiquitous and versatile sensor and interaction modules within signal transduction proteins. These PAS sensors can detect a diverse range of chemical and physical stimuli, consequently regulating the activity of various functionally diverse effector domains. Despite the extensive chemical, physical, and functional diversity associated with PAS sensors, the core structures of PAS domains remain broadly conserved [45]. AHR belongs to the distinctive bHLH/PAS protein family and is uniquely activated by small molecules, including various EDCs.

Protein–protein interactions occur within the PAS-A domain, which lacks a ligand-binding cavity. In contrast, the PAS-B domain forms a ligand-binding pocket (LBP) capable of accommodating diverse ligands. A distinctive feature of AHR is its versatile binding capacity to various ligands, including EDCs, phytochemicals, and endogenous metabolites [46,47,48]. A recent cryo-EM structural analysis of the indirubin-bound AHR complex unveiled the structural determinants of the PAS-B domain in promiscuous ligand binding. Notably, all secondary structures of the PAS-B domain, including a five-stranded antiparallel β-sheet (Aβ, Bβ, Gβ, Hβ, and Iβ) flanked by four consecutive α-helices (Cα, Dα, Eα, and Fα), contribute to the ligand binding pocket (LBP). This elongated channel—perpendicular to two helical structures—partially occupies the LBP, leaving a significant portion void, suggesting its capability to accommodate various small molecules of different sizes [49] (Figure 2B). A comparative study investigating the ligand-binding pockets (LBPs) of drosophila AHR (dAHR) and mouse AHR (mAHR) revealed that the larger size and structural variation of the mAHR PAS-B domain, which forms the LBP, contribute to its extensive ligand adaptability [50,51,52]. In contrast, dAHR possesses a smaller LBP and demonstrates constitutive activation in the absence of ligand binding, suggesting a species-dependent disparity in AHR actions [53].

Another notable aspect of AHR signaling involves the planar structure of AHR ligands. For instance, indirubin, characterized by its planar molecular structure and asymmetric double indole structure, intercalates between two layers of amino acid residues in the ligand-binding pocket (LBP). This characteristic underscores the crucial role of planarity in AHR ligands for selective ligand–AHR interactions [54,55,56,57]. Moreover, an interspecies difference exists in the binding affinity of AHR for the same ligand, despite a high level of sequence homology between mouse Ahr and human AHR (approximately 85%). For example, mouse AHR exhibits a ten-fold higher affinity for TCDD than human AHR [54,55,56]. In comparison, human AHR shows a much higher affinity for indirubin than mouse AHR, highlighting that variations in the ligand-binding pocket finely tune the specificity of ligand–AHR interactions despite its inherent promiscuity [57]. However, the ligand-dependent interactions between the PAS-A and PAS-B domains, which are likely responsible for recruiting different transcription factors and coregulators for activating a specific battery of genes, depending on ligand type (agonist vs. antagonist), remain to be further investigated.

## 4. EDCs from Environmental Pollutants and AHR

### 4.1. Dioxins and Dioxin-like Compounds

These compounds, identified and defined by the Stockholm Convention in 2001 as persistent organic pollutants (POPs), encompass polychlorinated dibenzofurans (PCDFs), polychlorinated dibenzo-p-dioxins (PCDDs), dibenzofurans (PCDFs), and polychlorinated biphenyls (PCBs). Recognizing their adverse effects on human health and the environment, the convention aimed to restrict their production. POPs, resistant to degradation through chemical, biological, and photolytic processes, are carbon-based substances. Their high stability and lipophilic nature lead to accumulation in the fatty tissues of humans and animals, causing adverse health effects. Beyond dioxin-like compounds, POPs encompass organochlorine pesticides (DDT, chlordane, dieldrin, heptachlor, hexachlorobenzene, mirex, and toxaphene) [58,59]. Accumulating evidence suggests that many POPs, including dioxins and dioxin-like compounds, exert biologically harmful effects by activating AHR function [60,61,62]. Dioxins and dioxin-like compounds are inadvertent byproducts resulting from high-temperature processes, including incomplete combustion of waste, coal, and wood, as well as automobile emissions. Industrial activities such as manufacturing chemicals, smelting, chlorine bleaching of paper pulp, and herbicide or pesticide production also contribute to their formation. The carbon skeleton of dioxins is represented by dibenzodioxin or dibenzo-p-dioxin. These compounds encompass polychlorinated dibenzo-p-dioxin (PCDD), polychlorinated dibenzofuran (PCDF) congeners, coplanar polychlorinated biphenyls (PCBs), and polybrominated biphenyl (PBB), which are bromine analogs of PCBs [63,64]. Natural disasters, including volcanic eruptions or forest fires, can also generate these toxic compounds [65]. Cigarette smoke contains elevated levels of dioxins or dioxin-like compounds.

TCDD, a prototypical dioxin and a most potent AHR agonist, exhibits a long half-life of 8 years in humans [66,67]. Short-term high exposure to dioxin can result in skin lesions, such as chloracne, and abnormal liver function [68,69,70]. Epidemiological studies have demonstrated that chronic dioxin exposure leads to impairments in the immune, nervous, cardiovascular, and reproductive systems and is associated with various types of cancers [61,71]. The International Agency for Research on Cancer (IARC) classifies dioxin as a Group 1 carcinogen [72]. The next section will describe the effects of the dioxin and dioxin-like compound-regulated AHR signaling axis on the pathogenesis of pancreatic diseases and cancer.

### 4.2. Polycyclic Aromatic Hydrocarbons (PAHs)

PAHs are organic compounds characterized by multiple aromatic rings. Exposure to PAHs can occur through various routes, including smoking, consumption of food and beverages, and inhalation of air. Major PAHs include benz[a]anthracene, chrysene, benzo[b]fluoranthene, and benzo[a]pyrene (BaP), with BaP being extensively studied for its carcinogenic and genotoxic properties [73,74]. Unlike persistent organic pollutants (POPs), PAHs have relatively short half-lives, ranging from 2.5 to 6.1 h [75,76]. The liver predominantly metabolizes PAHs through CYP enzymes, and the resulting metabolites are excreted in feces and urine. PAHs constitute a significant class of organic chemicals in particulate matter (PM) [77,78]. Multiple studies have robustly established an association between exposure to particulate matter (PM) and adverse human health effects, primarily attributed to the carcinogenic and mutagenic properties of polycyclic aromatic hydrocarbons (PAHs) [79,80]. Furthermore, many reports have underscored the correlation of PAH exposure with pancreatic diseases and cancer [81,82,83]. Nevertheless, the underlying mechanisms by which PAH exposure mediates the development of pancreatic diseases and cancer through the aryl hydrocarbon receptor (AHR) signaling pathway remain unexplored.

### 4.3. Hexachlorobenzene (HCB)

HCB is a chlorinated hydrocarbon that was historically employed as a fungicide or pesticide. Owing to its highly lipophilic nature, HCB is a pervasive pollutant that accumulates in biological systems and the environment [84]. Oral absorption represents a major route of HCB exposure. The accidental poisoning through HCB in Turkey from 1955–1959 highlighted the severe health consequences, with 4000 individuals exhibiting porphyria and skin lesions, and later developing arthritis [85]. In cell culture, HCB exposure has been shown to enhance cancer cell proliferation, migration, and invasion by activating AHR as a weak agonist [86]. Unlike polycyclic aromatic hydrocarbons (PAHs), HCB is classified as a weak AHR agonist and nongenotoxic carcinogen. Despite the fact that HCB exposure is a risk factor for the development of pancreatic diseases and cancer [87,88,89], the role of the HCB-AHR signaling axis remains uninvestigated.

### 4.4. Bisphenol A (BPA)

BPA is a chemical that produces various polycarbonate plastics, including food containers, baby bottles, water bottles, medical devices, and hygiene products. It has been a widely used endocrine-disrupting chemical (EDC) since the early 1950s. It is a major constituent of polycarbonate plastics used in manufacturing epoxy resins, dental sealants, and recycled paper, and is used in the lining of food cans [90,91]. BPA is detectable in urine, blood, breast milk, and other tissues, with the primary human exposure route being ingestion. Upon ingestion, BPA is rapidly absorbed and metabolized in the liver, becoming hydrophilic and subsequently excreted primarily in urine, with a known half-life of less than six hours [92,93]. Despite being a non-persistent EDC with a short half-life, over 90% of individuals exhibit detectable urine BPA levels. Many studies have reported a positive correlation between urine BPA levels and diabetes in adults and children [94,95]. Many studies have also reported that BPA exposure disrupts pancreatic β-cell function by targeting estrogen receptors α or β [96,97,98,99,100]. Additionally, BPA activates aryl hydrocarbon receptor (AHR) signaling and inhibits mouse ovarian follicle growth [101]. Exposure to BPA during mouse embryo development increases the expression of AHR and its downstream target genes [102]. Additionally, low-dose BPA exposure activates AHR in breast cancer cells, increasing their aggressive cancer cell phenotype [103]. However, the effects of BPA on AHR signaling involving development of pancreatic diseases and cancer remain to be further investigated. A recent report demonstrated that exposure to bisphenol A (BPA) in a mouse model activated the aryl hydrocarbon receptor (AHR) in pancreatic islets, indicating a potentially harmful role of BPA-AHR signaling activation in the pancreas [104]. Nevertheless, further mechanistic studies investigating the role of BPA-regulated AHR pathways in the pathogenesis of pancreatic diseases and cancer are warranted.

### 4.5. Heavy Metals

Exposure to heavy metals, such as arsenic, cadmium, or chromium, can result in cellular injury, genetic alterations, or a combination of both [105,106]. Arsenic, a prevalent toxic metal in the environment, induces the generation of reactive oxygen species (ROS) upon exposure, disrupting antioxidant defense mechanisms and impacting mitochondrial morphology and integrity. Chromium, particularly in its hexavalent form [Cr(VI)], is highly toxic, mutagenic, and carcinogenic, producing hydroxy radicals and superoxide, contributing to adverse effects. Cadmium, identified as a human carcinogen, also induces oxidative stress. The toxicity and carcinogenicity associated with heavy metals often involve the production of ROS [107,108]. ROS generated by heavy metal exposure likely indirectly activate AHR signaling. Exposure to arsenic or cadmium alone increases AHR activity and the expression of downstream target genes, such as CYP1A1 [109,110], and is possibly linked to the generation of oxidative stress that increases the production of biliverdin or bilirubin, an AHR ligand [111]. Similarly, oxidative stress induced by chromium (VI) exposure activates AHR signaling by increasing the production of oxindole, an AHR ligand. Cadmium exposure also elevates AHR signaling and downstream gene expression [112]. Moreover, heavy metal exposure and AHR ligand treatment further enhance AHR signaling activation [113,114]. These reports emphasize the pivotal role of oxidative stress in heavy metal exposure-regulated aryl hydrocarbon receptor (AHR) actions. Many studies have identified heavy metal exposure as a significant risk factor for pancreatic cancer [115,116,117]. However, the underlying mechanisms connecting heavy metal exposure-mediated AHR signaling regulation to the development of pancreatic diseases and cancer remain unknown.

Collectively, we summarize the interplay between endocrine-disrupting chemicals (EDCs) and aryl hydrocarbon receptor (AHR) signaling, as well as epidemiological studies and relevant findings on the roles of these EDC-mediated AHR signaling regulations (Table 1).

## 5. Roles of EDC–AHR Interactions in the Pathogenesis of Pancreatic Diseases and Cancer

In its multifunctional role, the pancreas plays a crucial part in the endocrine and exocrine systems. Regarding endocrine function, the islets of Langerhans within the pancreas consist of five distinct cell types—alpha, beta, delta, epsilon, and upsilon—each responsible for secreting specific hormones. These hormones include glucagon, insulin, somatostatin, ghrelin, and pancreatic polypeptide. On the other hand, exocrine function involves acinar cells releasing digestive enzymes such as amylase, lipase, and proteases. These enzymes are channeled into the pancreatic duct, which merges with the common bile duct, and their combined secretions enter the duodenum, aiding in the breakdown of carbohydrates, fats, and proteins from ingested food.

Exposure to EDC can have adverse effects on the pancreas, contributing to conditions such as obesity, diabetes, insulin resistance, hyperinsulinemia, and pancreatitis. Importantly, these conditions also serve as known risk factors for the development of pancreatic cancer [118,119]. Understanding the impact of EDC exposure on the intricate functions of the pancreas is essential for comprehending the potential health risks associated with these environmental contaminants.

### 5.1. Role of EDC-Regulated AHR in Diabetes Mellitus

Diabetes mellitus (DM) is a chronic metabolic disorder marked by elevated blood glucose levels due to compromised insulin secretion and disrupted glucose homeostasis. According to the IDF Diabetes Atlas, 537 million adults aged 20–79 had diabetes globally in 2021. Projections indicate an increase to 643 million by 2030 and 783 million by 2045 [120]. Two primary types of DM exist: type 1 DM (T1DM), characterized by total insulin absence, frequently found in children; and type 2 DM (T2DM), primarily found in adults, resulting from diminished insulin secretion and functionality.

#### 5.1.1. Type 1 Diabetes Mellitus (T1DM)

T1DM is an autoimmune disorder characterized by the destruction of pancreatic β cells by self-reactive T cells, resulting in insulin deficiency. The incidence of T1DM has markedly increased over past decades, and this surge cannot be exclusively attributed to genetic factors. A potential contributing factor is the heightened exposure to endocrine-disrupting chemicals (EDCs) during prenatal and early developmental stages. This exposure may disrupt immune homeostasis, which regulates the maintenance and survival of pancreatic β cells. Numerous epidemiological studies suggest that exposure to environmental contaminants, such as dioxins, PCBs, bisphenol A, and air pollutants containing PAHs, increases the risk of type 1 diabetes mellitus (T1DM) [121,122].

AHR is expressed in various immune cell types, including dendritic and T cells, with T cells playing a pivotal role in destroying pancreatic β cells [123]. In the non-obese diabetic (NOD) mouse model of type 1 diabetes mellitus (T1DM) development, TCDD activated AHR, increasing Foxp3+ T cells that exert anti-inflammatory effects against effector T cells, preventing T1DM development [124]. In the same mouse model, AHR activation by the exogenous ligand 10-CI-BBQ inhibited T1DM development [125]. These findings suggest that AHR signaling activation, depending on the immune cell type, is immunosuppressive and can modulate immune responses during T1DM development, highlighting the potential of AHR targeting in T1DM. In contrast to the role of AHR in T1DM, a recent study reported that high levels of urinary metabolites of polycyclic aromatic hydrocarbons (PAHs) are associated with an increased risk of T1DM in children and adolescents [126]. In animal studies, prenatal exposure to a mixture of PAHs resulted in dysfunctional pancreatic islets associated with T1DM [127]. PAHs may increase the risk of T1DM development through other unknown mechanisms.

High urinary levels of bisphenol A (BPA) in children and adolescents are associated with increased T1DM development [128]. In animal studies, transmaternal exposure to BPA significantly increases insulitis severity and diabetes incidence in female offspring in a dose-dependent manner [129]. The progeny of BPA-exposed mothers exhibit heightened apoptosis of both pancreatic α and β cells, promoting the development of T1DM in NOD mice [130]. In another streptozotocin (STZ)-induced T1DM murine model, low-dose multiple oral BPA exposures facilitated diabetes induction, potentially through BPA-mediated immunomodulation of T cells or reduced cytokine levels, suggesting that BPA acts as a risk factor for diabetes by altering immune modulatory activity [131]. A recent report showed that BPA-activated AHR in pancreatic islets disrupted glucose homeostasis and altered insulin sensitivity, implying that BPA-mediated AHR signaling activation plays a role in T1D development [104]. Epidemiological evidence and molecular studies linking BPA exposure, AHR, and T1DM development need further investigation.

Elevated levels of arsenic and fluoride in drinking water have been associated with an increased incidence of type 1 diabetes mellitus (T1DM) [132]. The high plasma level of arsenic is associated with an increased risk of T1DM development [133]. Low-level sub-chronic arsenic exposure from the prenatal stage has been shown to impair glucose metabolism in adult life [134]. It has been previously reported that arsenic and other heavy metals regulate AHR signaling [109,110,111,112,113,114]. However, the effects of arsenic exposure on AHR activity in the pancreas associated with T1DM development remain unclear. Further human studies and longitudinal evidence are needed to explore this relationship.

#### 5.1.2. Type 2 Diabetes Mellitus (T2DM)

Many epidemiological studies have consistently indicated a connection between dioxin exposure and the development of type 2 diabetes mellitus (T2DM). American war veterans who engaged in defoliant spraying containing TCDD during the Vietnam War showed a correlation between their TCDD exposure levels and the incidence of T2DM [135]. A significant dose–response relationship was observed between the serum concentration of six selected persistent organic pollutants (POPs)—including dioxin and dioxin-like compounds—and the prevalence of diabetes [136]. Furthermore, prolonged exposure to persistent organic pollutants (POPs), including polychlorinated biphenyls (PCBs) and organochlorine pesticides, was associated with lower serum insulin levels. Correspondingly, in vitro studies consistently demonstrated impaired insulin secretion in pancreatic β-cells following low-level POP exposure [137].

Mounting evidence suggests that exposure to TCDD targets pancreatic β-cells, disrupting insulin secretion regulation. Isolated pancreatic islet cells from rats exposed to low chronic TCDD exhibited impaired glucose-stimulated insulin secretion [138]. Similarly, low TCDD exposure to the insulin-secreting pancreatic β-cell line INS-1E significantly impaired insulin secretion, accompanied by increased pancreatic β-cell death, implicating the sensitivity of pancreatic β-cells to dioxin exposure [139]. TCDD exposure to human embryonic stem cells compromised pancreatic lineage differentiation and altered DNA methylation and gene expression, highlighting that early embryonic TCDD exposure dysregulates pancreatic function and increases the risk of type 2 diabetes mellitus (T2DM) [140]. Furthermore, several animal studies revealed that TCDD-activated AHR reduced insulin secretion. Exposure to TCDD suppressed insulin secretion and increased pancreatic β cell death [141]. Intriguingly, in CYP1A1 and CYP1A2 knockout islets, the toxic effects of TCDD or 3-MC were decreased, emphasizing the crucial role of CYP1 enzyme activities in pancreatic beta cell survival and death [142]. In an experimental model of T2DM induced by streptozotocin (STZ), significantly elevated CYP1A1 activity levels were observed in diabetic rats, supporting the role of the AHR-CYP1 axis in pancreatic β-cell pathophysiology [143]. Moreover, the crucial role of AHR in TCDD-mediated toxic effects on insulin secretion and glucose homeostasis was emphasized through the use of AHR knockout mice, which demonstrated imbalanced glucose homeostasis and reduced insulin levels [144]. A single instance of acute dioxin exposure to mice suppressed insulin secretion for up to 6 weeks, suggesting that the toxic effects of acute dioxin exposure on pancreatic β-cells can have long-term consequences, even if the exposure is transient [145]. Low levels of POP exposure, including organochlorine pesticides and PCBs, decreased insulin secretion by disrupting pancreatic β cell function [146].

Exposure to BPA, PAHs, hexachlorobenzene, or heavy metals has been acknowledged as a risk factor for the development of T2DM [147,148,149,150,151]. As detailed in the preceding sections, these EDCs act as positive or negative AHR signaling regulators. Nevertheless, the underlying mechanisms through which EDC exposure modulates AHR signaling in the context of T2DM development remain to be further investigated.

#### 5.1.3. Role of EDC-Regulated AHR in Pancreatitis

Pancreatitis is an inflammatory disease of the pancreas, accompanied by the gradual replacement of the pancreas by fibrotic tissue compartments. Heavy alcohol consumption and cigarette smoke are major risk factors of pancreatitis [152]. Notably, cigarette smoke contains high levels of dioxins—dioxin-like compounds that activate the AHR signaling pathway [153,154]. Pancreatitis is a key risk factor for pancreatic cancer development [155,156].

Acute pancreatitis, characterized by inflammatory damage to the pancreatic acini, leads to extensive necrosis and multi-organ failure, contributing to severe cases’ mortality. The aryl hydrocarbon receptor (AHR) emerges as a critical transcription factor pivotal for the production of IL22, a protective cytokine in acute pancreatitis. In a murine model induced by caerulein, AHR inactivation using the antagonist CH223191 decreased IL22 production. At the same time, AHR activation by biliverdin, an AHR agonist, increased pancreatic IL22 levels and provided protection against acute pancreatitis [157]. Conversely, acute exposure to benzo(a)pyrene or TCDD induces pancreatitis accompanied by oxidative stress-related mitochondrial respiratory dysfunction. Resveratrol, an AHR antagonist, prevents the harmful effects of pancreatitis and mitigates mitochondrial damage [158]. TCDD exposure also upregulates long noncoding RNA MALAT1 expression in pancreatic cancer cells and tissue. MALAT1 interacts with the histone methyltransferase EZH2, enhancing its enzymatic activity. Consequently, AHR-mediated MALAT1 induction amplifies EZH2’s histone methyltransferase activity, revealing a novel pathway through which TCDD exposure alters epigenetic status via activation of the AHR-MALAT1-EZH2 signaling axis [159]. In experimental autoimmune pancreatitis, AHR activation augments IL22 production in pancreatic α cells, suppressing chronic fibrotic and inflammatory processes via IL22 production [160].

Chronic pancreatitis is a progressive inflammatory condition characterized by increased fibrosis, serving as a predisposing factor for pancreatic cancer. Cigarette smoke—a significant risk factor for chronic pancreatitis—contains elevated levels of dioxin-like compounds that activate aryl hydrocarbon receptor (AHR) signaling. In a murine model of chronic pancreatitis, exposure to cigarette smoke resulted in increased IL22 production in T cells, promoting pancreatic fibrosis and contributing to the development of chronic pancreatitis. Consistently, cigarette smokers in this model exhibited higher serum levels of IL22 than non-smokers [161]. These reports suggest that the context-dependent effects of AHR activation, influenced by the disease model or AHR ligand type, add complexity to the nature of AHR signaling.

#### 5.1.4. Role of EDC-Regulated AHR in Pancreatic Cancer

Pancreatic cancer stands as the seventh leading cause of global cancer-related mortality, characterized by its formidable nature, with symptoms often emerging only at an advanced stage, resulting in elevated mortality rates. In 2018, approximately 450,000 new cases of pancreatic cancer were reported worldwide, leading to 432,242 deaths [162]. In the United States, pancreatic cancer accounts for 3% of overall cancer incidence and contributes to 7% of cancer-related deaths. Projections for 2023 estimate 64,050 new cases with 50,550 fatalities. The five-year survival rate for pancreatic cancer is a mere 12%, emphasizing the challenge posed by late-stage diagnoses, with only 12% identified at an early, surgically removable stage. Over 50% of individuals receive diagnoses at later stages, characterized by distal metastasis. Current therapeutic modalities lack efficacy for advanced stages, and options for efficacious early diagnosis remain limited [163,164]. Pancreatic ductal epithelial cells, among the different pancreatic cell types, are the origin of pancreatic adenocarcinomas, a major type of pancreatic cancer.

Epidemiological studies have indicated a moderate increase in the risk of pancreatic cancer associated with high serum concentrations of persistent organic pollutants (POPs) [165,166]. However, further research is needed to substantiate these findings. Notably, cigarette smoking stands out as a significant risk factor for pancreatic cancer, with high levels of dioxin-like compounds present in cigarette smoke [153]. Cigarette smoke has been shown to induce aryl hydrocarbon receptor (AHR) activation [154]. These observations highlight a potential molecular link between dioxin exposure, AHR activation, and the development of pancreatic cancer, prompting further investigation.

Using rats as a model, it was found that chronic exposure of dioxin or dioxin-like compounds in the pancreas increased cytoplasmic vacuolation, inflammation, and atrophy in the exocrine pancreas, accompanied with low incidence of pancreatic acini adenoma and carcinoma, indicating that pancreatic acini is a target tissue of dioxin and dioxin-like compounds [167]. AHR displays heightened expression in the cytoplasm of pancreatic cancer tissues, and its activation by AHR agonists such as DIM (diindolymethane) inhibits the growth of pancreatic cancer cells [168]. Similarly, omeprazole, functioning as an AHR agonist, hampers the migration and invasion of pancreatic cancer cells [169]. Carbidopa, an FDA-approved drug for Parkinson’s disease, acts as an AHR agonist, impeding the growth of pancreatic cancer cells by inhibiting IDO1 (indoleamine 2,3-dioxygenase-1) [170]. Depletion of AHR using small interfering RNAs targeting AHR in pancreatic cancer cells heightens sensitivity to gemcitabine, a chemotherapeutic agent, and diminishes cells’ invasive and migratory potential [171]. However, the effects of ligand-dependent AHR activation or inhibition on pancreatic cancer cell proliferation, invasion, and migration have not been investigated. Oct4, a master transcription factor of pluripotency that mediates cancer stem cell features, is suppressed by the tryptophan-derived AHR ligand ITE. ITE interacts with AHR and suppresses Oct4 expression, directing cancer stemness and growth [172]. These reports introduced additional layers of complexity in AHR activities; the type of ligand appears to dictate the regulation of distinct gene subsets, potentially encompassing both pro-oncogenic and tumor-suppressive genes.

AHR also functions as a sensor of microbiome-derived metabolites, modulating immune function within the tumor microenvironment. Notably, tumor-associated macrophages (TAMs) express high levels of AHR. The depletion of AHR in myeloid cells or the inhibition of AHR by antagonistic treatment reduces the progression of pancreatic ductal adenocarcinoma (PDAC) by diminishing the immunosuppressive function of TAMs and enhancing immune surveillance by CD8+ T cells. Correspondingly, high AHR expression in PDAC patients correlates with poor clinical outcomes and features of immunosuppressive TAMs, supporting the tumor-promoting role of AHR in PDAC [173]. These findings underscore the immunomodulatory functions of AHR in TAMs while emphasizing the necessity for further investigations into the interactions between immune cell types and ductal epithelial cells during the development of pancreatic cancer. The role of AHR as a molecular interface between EDC exposure and the development of various pancreatic diseases, including cancer, is summarized below (Figure 3).

In summary, numerous epidemiological studies substantiate a positive correlation between exposure to various EDCs and the incidence of diverse pancreatic diseases, including cancer. Mounting evidence suggests that many EDCs activate or interfere with aryl hydrocarbon receptor (AHR) signaling. However, there is a gap in our understanding regarding AHR’s molecular and mechanistic roles in the EDC-mediated development of various pancreatic diseases and cancer. Moreover, the intricate nature of AHR signaling, manifesting as activation or inhibition depending on ligand type, tissue/cell context, or disease model, poses a major obstacle in investigating the Janus-like role of AHR in promoting or inhibiting pancreatic pathogenesis. Despite these challenges, the ligand-dependent activation or inhibition of AHR’s actions by selective AHR modulators (SAhRMS) underscore its potential as a promising molecular target [174].

## Figures and Tables

**Figure 1 ijms-25-03818-f001:**
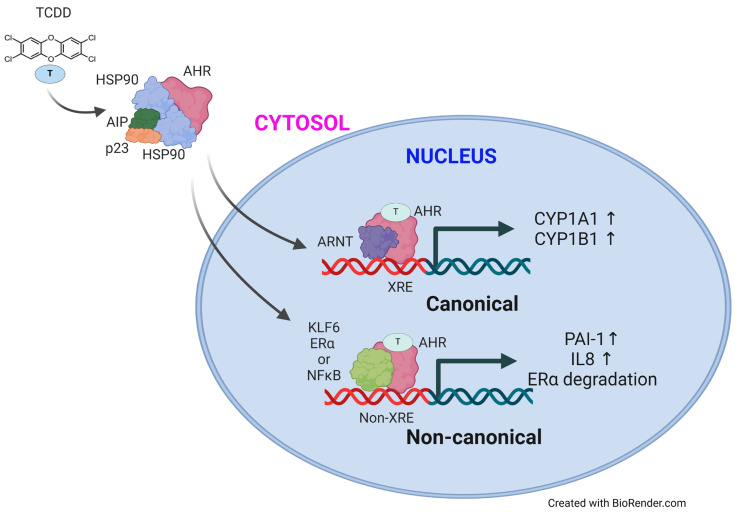
Activation of AHR signaling in canonical and non-canonical manners. Upon ligand binding, AHR translocates into the nucleus in the canonical pathway, where it forms a heterodimer with ARNT. This complex then interacts with Xenobiotic Response Elements (XRE) on target gene promoters, initiating downstream gene transcription. Conversely, in the non-canonical pathway, ligand-bound AHR interacts with alternative transcription factors such as KLF6, ERα, and NFκB, leading to the activation of gene transcription or degradation of transcription factors independent of XRE interaction.

**Figure 2 ijms-25-03818-f002:**
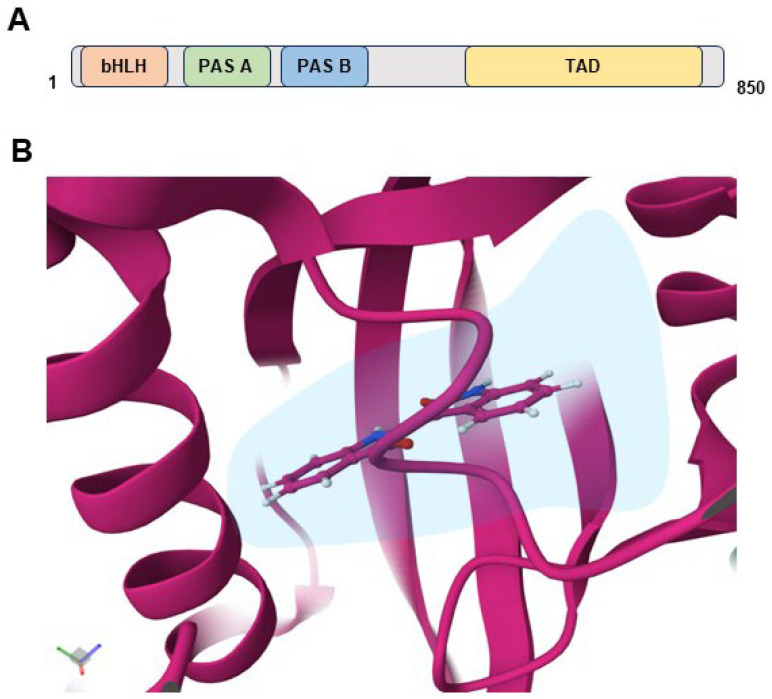
AHR structure and ligand binding pocket (LBP). The domain structure of human AHR contains bHLH, PAS A, PASB, and TAD, totaling 850 amino acids. (**A**) A close-up view of the ligand binding pocket (LBP in magenta color) shows a transparent surface with the ligand indirubin (middle). The AHR LBP interacts with the AHR agonist indirubin (**B**).

**Figure 3 ijms-25-03818-f003:**
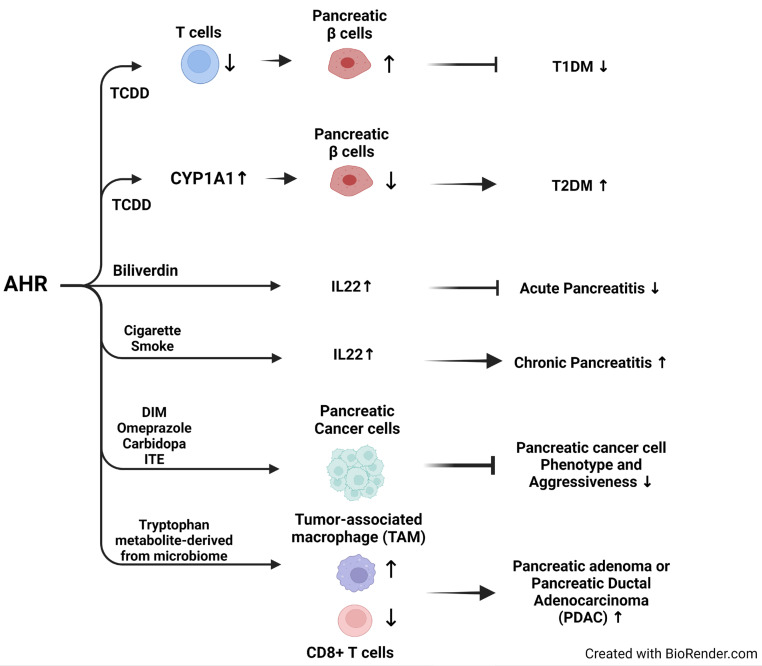
Role of AHR in EDC-mediated pancreatic pathogenesis. The references related to T1DM are [124,125,126,127], T2DM [138,139,140,141,142,143,144,145,146], acute pancreatitis [157,158,159,160], chronic pancreatitis [161], and pancreatic cancer [167,168,169,170,171,172,173].

**Table 1 ijms-25-03818-t001:** A summary of the interplays between endocrine-disrupting chemicals (EDCs), AHR signaling, epidemiological reports, and relevant findings on the roles of these EDC-mediated AHR signaling regulations in the development of pancreatic diseases and cancer. References are indicated as [ ].

EDC	Regulation of AHR Signaling	Epidemiological Studies Relevant to Pancreatic Diseases or Cancer	Mechanistic Role of AHR in Pancreatic Diseases and Cancer
Dioxin and dioxin-like compounds	AHR agonists [60,61,62]	See 5. Roles of EDC–AHR Interactions in the Pathogenesis of Pancreatic Diseases and Cancer	See 5. Roles of EDC–AHR Interactions in the Pathogenesis of Pancreatic Diseases and Cancer
Polycyclic aromatic hydrocarbons	AHR agonists and oxidative stress inducers [73,74,75,76,77,78,79,80]	[81,82,83]	Unknown
Hexachlorobenzene	Weak AHR agonist [86]	[87,88,89]	Unknown
Bisphenol A	Weak AHR agonist [101,102,103,104]	[94,95,96,97,98,99,100]	Unknown
Heavy metals	AHR agonists and oxidative stress inducers [109,110,111,112,113,114]	[115,116,117]	Unknown

## Data Availability

No new data were created or analyzed in this study. Data sharing is not applicable to this article.

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
