# Peer review of "The Role of Endocrine Disruption Chemical-Regulated Aryl Hydrocarbon Receptor Activity in the Pathogenesis of Pancreatic Diseases and Cancer"

_ijms, 2024, doi:10.3390/ijms25073818_

Round 1

Reviewer 1 Report

Comments and Suggestions for Authors

Kyounghyun Kim submitted a manuscript entitled “Role of Endocrine Disruption Chemicals-Regulated Aryl Hydrocarbon Receptor Activity in Pancreatic Pathogenesis”.

This review focuses on current knowledge of endocrine-disruptor chemicals as  agonists or antagonists of AHR and their roles in the pathogenesis of pancreatic diseases, including cancer.

In general, the manuscript is well written. But there are some comments to the text of the manuscript that should be taken into account.

 1.      Pathogenesis is the mechanism of the occurrence and development of diseases, but not an organ. Therefore, the phrase "pancreatic pathogenesis" should be replaced by "pathogenesis of pancreatic disease" in the lines 3, 15, 17, 231, 434, 444.

 2.      Lines 29-32. It is necessary to make references to the EDCs spectrum.

 3.      Lines 136. In the title of the chapter 4, it should be clarified that “EDCs from environmental pollutants", since only these EDCs are considered in the chapter.

4.      The list of abbreviations should include T2DM, T1DM, DIM, TFMs.

Author Response

Thank you for dedicating time to review our manuscript. Below, we provide detailed responses to the comments, and the corresponding revisions are highlighted in the resubmitted files.

  1. Pathogenesis is the mechanism of the occurrence and development of diseases, but not an organ. Therefore, the phrase "pancreatic pathogenesis" should be replaced by "pathogenesis of pancreatic disease" in the lines 3, 15, 17, 231, 434, 444.

RESPONSE: Thanks for pointing these out. Now, all the phrases are replaced.

  1. Lines 29-32. It is necessary to make references to the EDCs spectrum.

RESPONSE: We entirely agree with the reviewer’s perspective. We make all the necessary references to the EDCs.

  1. Lines 136. In the title of the chapter 4, it should be clarified that “EDCs from environmental pollutants", since only these EDCs are considered in the chapter.

RESPONSE: Thanks for pointing this out. This part is corrected as suggested.  

  1. The list of abbreviations should include T2DM, T1DM, DIM, TAMs.

RESPONSE: Thank you for bringing this to our attention. We include these in the abbreviation.   

Reviewer 2 Report

Comments and Suggestions for Authors

The author has chosen an interesting topic and attempted to review the current literature extensively. The present manuscript, however, could be improved if it took into account the following suggestions:

1- include a Materials and Methods section indicating the keywords and kind of references and the search criteria adopted for the review: those wishing to repeat the search should be able to do so and obtain the same scientific work

2- specify well, where HIF-1a is cited (line 63), the difference with ARNT. It appears from the text that they are the same thing, but that is not true

3-In the intracellular complex of AHR, another factor is also involved in regulation: AHRR. It would be helpful to also make explicit the role of AHRR, if any, within the AHR system and the different pancreatic diseases in which AHR is involved

4-The chapter dealing with the in-depth description of disruptors (4. EDCs and AHR) is too long and perhaps unnecessary: the main purpose of the paper is to see AHR in pancreatic pathology and not the inducers of AHR. This part could be summarized in a table schematically summarizing what is written there.

Comments on the Quality of English Language

only modest English revision required

Author Response

Thank you for dedicating time to review our manuscript. Below, we provide detailed responses to the comments, and the corresponding revisions are highlighted in the resubmitted files.

1- include a Materials and Methods section indicating the keywords and kind of references and the search criteria adopted for the review: those wishing to repeat the search should be able to do so and obtain the same scientific work

RESPONSE: We agree with the reviewer’s perspective. We provided keywords as search criteria to promote literature search related to this topic.   

2- specify well, where HIF-1a is cited (line 63), the difference with ARNT. It appears from the text that they are the same thing, but that is not true.

RESPONSE: Thanks for pointing this out. We correct it.

3-In the intracellular complex of AHR, another factor is also involved in regulation: AHRR. It would be helpful to also make explicit the role of AHRR, if any, within the AHR system and the different pancreatic diseases in which AHR is involved.

RESPONSE: We fully align with the reviewer's viewpoint.  We incorporate the description of the role of AHRR as a negative feedback repressor of the AHR signaling activation in section 2 of  Aryl Hydrocarbon Receptor” [40]. However, almost no reports indicate a role of AHRR in the pathogenesis of pancreatic diseases and cancer, except one report showing that cigarette smoke exposure induced AHRR expression in the murine model of pancreatitis [160].

4-The chapter dealing with the in-depth description of disruptors (4. EDCs and AHR) is too long and perhaps unnecessary: the main purpose of the paper is to see AHR in pancreatic pathology and not the inducers of AHR. This part could be summarized in a table schematically summarizing what is written there.

RESPONSE: We concur with the reviewer's assessment and have addressed this concern by incorporating Table 1. This table presents a comprehensive overview, elucidating the correlation between endocrine-disrupting chemicals (EDCs), Aryl hydrocarbon receptor (AHR) signaling, epidemiological reports, and pertinent findings on the involvement of EDC-mediated AHR signaling regulations in the progression of pancreatic diseases and cancer.